# Tensile Behavior of Basalt Textile Reinforced Concrete: Effect of Test Setups and Textile Ratios

**DOI:** 10.3390/ma15248975

**Published:** 2022-12-15

**Authors:** Chenglin Wan, Jiyang Wang, Shubin Wang, Xiaohua Ji, Yu Peng, Hongmei Zhang

**Affiliations:** 1Institute of Engineering Mechanics, China Earthquake Administration, Harbin 150080, China; 2College of Civil Engineering and Architecture, Zhejiang University, Zijingang Campus, Hangzhou 310058, China; 3Center for Balance Architecture, Zhejiang University, Hangzhou 310007, China; 4The Architectural Design & Research Institute of Zhejiang University Co., Ltd., Hangzhou 310007, China

**Keywords:** basalt textile reinforced concrete, test setup, textile ratio, tensile behavior

## Abstract

The clevis-grip tensile test is usually employed to evaluate the mechanical properties of textile reinforced concrete (TRC) composites, which is actually a bond test and is unsuitable for determining reliable design parameters. Thus, the clevis-grip tensile test needs further improvement to obtain foreseeable results concerning TRC tensile behavior. This paper presents the experimental results of twenty-one tension tests performed on basalt TRC (BTRC) thin plates with different test setups, i.e., clevis-grip and improved clevis-grip, and with different textile ratios. The influences of test setups and textile ratios on crack patterns, failure mode, and tensile stress-strain curves with characteristic parameters were analyzed in depth to judge the feasibility of the new test setup. The results indicated that with the new test setup, BTRC composites exhibited textile rupture at failure; in addition, multi-cracks occurred to the BTRC composites as the textile ratio exceeded 1.44%. In this case, the obtained results relied on textile properties, which can be considered reliable for design purposes. The modified ACK model with a textile utilization rate of 50% provided accurate predictions for the tensile stress-strain behavior of the BTRC composite derived from the improved test setup. The proposed test setup enables the adequate utilization of BTRC composite and the reliability of obtained results related to the occurrence of textile rupture; nevertheless, further work is required to better understand the key parameters affecting the textile utilization rate, such as the strength of the concrete matrix.

## 1. Introduction

Textile reinforced concrete (TRC) composites appear as a competitive alternative to fiber reinforced polymer (FRP) composites for strengthening existing concrete or masonry structures [1,2,3,4,5]. Compared to FRP, the main advantages of TRC come from the use of an inorganic matrix, i.e., fine-grained concrete, which offers the possibility to overcome some drawbacks caused by using epoxy resin, such as low thermal (fire) resistance [6], inapplicability on wet surfaces [7], high irreversibility [8], and low compatibility with the inorganic substrate [1]. TRC comprises fibers arranged in the form of multi-axial textile impregnated by and embedded in fine-grained concrete. Externally bonded TRC is conventionally utilized in structural reinforcement scenarios, and the strengthening efficiency depends on the TRC-substrate interface capacity and tensile properties of the TRC composite [9]. The tensile behavior of TRC composite has been widely studied at different dimensions and using different test setups, which is influenced by test setups [10,11,12,13], textile ratios [14], mechanical properties of matrix and fiber yarns, and the interface properties between concrete matrix and fiber yarns [15,16].

Recently, both the Italian CNR-DT 215/2018 [17] and US ACI 549.6R-20 [18] guidelines have proposed test setups to assess the tensile performance of the TRC system for structural design. The two guidelines apply different tensile test setups distinguished by the gripping systems of TRC specimens, leading to significant differences in the obtained results even for the same specimens [12]. Particularly, in the ACI 549.6R-20 document, the tensile force is applied to the TRC matrix through a clevis-type grip device, which transfers the force from metallic plates to the matrix by shearing stress to realize the loading of the whole TRC composite (Figure 1a). Significant slippage between the textile and matrix always occurs in the test setup shown in Figure 1a, and the typical failure of TRC composite is not the textile rupture [11,19]. The tensile performance of the TRC composite is extremely poor, very scatted, and hardly predicted [20]. The response of the TRC composite under tension is highly dependent on the metallic plates-TRC interface bonding properties, and the results of clevis-grip tensile tests show strong dependencies on factors other than the material properties, such as the length of the gripping device [20]. Using the parameters of the strengthening system determined by the clevis-grip tensile test to design TRC-strengthened structures might be questionable. According to the Italian acceptance criteria [21], the application of tensile force is realized by exerting transversal compressive stress (clamping-grip device, Figure 1b) to reduce/prevent the slippage of the textile within the matrix in the extremities of the TRC coupon. Clamping-grip tensile tests are used to determine the overall tensile response of the TRC composite [22,23,24]. The failure of TRC composite usually takes place indicated by the rupture of textile reinforcement [25]. TRC composite exhibits high strength and large deformability. In this case, the tensile stress-strain behavior of the TRC composite is relatively stable and commonly characterized by a trilinear curve, including the initial elastic stage, crack-development stage, and crack-widening stage. The ACK model proposed by Aveston and Kelly [26] and Cuypers and Wastiels [27] is always adopted to predict the trilinear tensile behavior of TRC. However, the influence of the transversal compressive stress (applied to the gripped parts of the specimen) on the interactions between metallic plates, matrix, and textile remains unclear, and the threshold of the transversal compressive stress making the slippage negligible, is unknown. As considerable textile–matrix slippage occurs, the mechanical properties derived from the clamping-grip tests are still significantly dependent on unexpected parameters, such as the gripping length and the transversal compressive stress level, rather than the material properties. It is difficult to obtain reliable mechanical properties of the TRC. In general, the two widely used tensile tests (Figure 1a,b) seem unsuitable for assessing the tensile properties of TRC composites. Developing new test setups to obtain the reliable mechanical properties of TRC for structural design is of significant importance.

Similar to that observed in the clevis-grip tensile test, the typical failure of TRC composite directly applied in structures is considerable slippage of textile with respect to the surrounding matrix [28]. As a result, the TRC composite is far from fully utilized. Therefore, establishing a new tensile test setup for TRC should consider the following aspects: (1) the acquired tensile properties of TRC composite are taken full advantage of and are reliable for design purposes; (2) corresponding constructional measures in structural reinforcement are easy to accomplish. Motivated by this, the authors improve the clevis-grip tensile test by casting the end zone of the TRC composite with epoxy resin to enforce the failure mode of TRC as the textile fracture to obtain outstanding and reliable testing results (Figure 1c). Notably, difficulties always arise from the quantitative analysis of the effect of transverse compressive stress on the textile–matrix shear stress transfer mechanism and from the determination of the transverse compressive stress threshold. Thus, the clamping grip tensile test that may obtain unreliable results is eliminated herein.

This paper compared the mechanical properties of basalt TRC (BTRC) composite obtained by different tensile test setups, i.e., clevis-grip tensile test, and its improved form to determine appropriate test setups for characterizing design parameters of TRC composite. In addition, the effect of textile ratios on the tensile behavior of TRC was investigated to provide instructive suggestions for TRC engineering applications. Twenty-one TRC coupons with different textile ratios and casting procedures were prepared for uniaxial tension tests. The experimental results were discussed in terms of crack patterns, failure modes, and tensile stress-strain behavior with some extracted parameters. Finally, the ACK model was modified by considering the textile utilization rate to predict the tensile stress-strain curve of the BTRC composite, enabling the design of TRC-strengthened structures with improved performance.

## 2. Experimental Program

### 2.1. Materials

The studied BTRC composite consisted of basalt textile and fine-grained concrete. The fine-grained concrete was mixed with P·O 52.5 ordinary Portland cement, fine sand, silica fume, grade II fly ash, superplasticizer, and water. Table 1 shows the detailed mix proportions. According to the standard EN 1015-11 [29], six 40 mm × 40 mm × 160 mm fine-grained concrete prisms were cast to determine the 28-day compressive and flexural strengths as 58.1 MPa [Coefficient of variation (CoV) = 5.0%)] and 5.6 MPa (CoV = 4.0%), respectively. CoV is defined as the ratio of the standard deviation to the mean. Following ASTM C469 [30], nine compression tests were performed on cylinders (diameter = 100 mm, height = 200 mm) to measure the elastic modulus of fine-grained concrete as 32.0 GPa (CoV = 8.8%).

The basalt textile was weaved of untwisted basalt fiber yarns with equal spacing of 25 mm in the two orthogonal directions, as shown in Figure 2a. The equivalent textile thickness was 0.12 mm, and the areal density, i.e., the mass per unit area, of basalt textile was 353 g/m^2^. Each fiber yarn was composed of hundreds of filaments with diameters of roughly 10 μm (Figure 2b). Five basalt fiber yarns with a length of 400 mm were prepared and tested under uniaxial tension at a loading rate of 2 mm/min. The experimental stress-strain curves had a linear branch up to the peak points, followed by an abrupt drop related to textile rupture. The basalt fiber yarns showed an average tensile strength of 493 MPa (CoV = 11.4%), an average maximum strain of 1.8% (CoV = 11.5%), and an average elastic modulus of 27.8 GPa (CoV = 3.6%).

### 2.2. Specimen Preparation

The influences of test setups and textile ratios on the tensile behavior of BTRC composite were explored in this study. The adopted test setups included clevis-grip and improved clevis-grip realized by changing the specimen fabrication process (Figure 3b,c). The textile ratio was defined as the ratio of the cross-sectional area of the fiber yarns to that of the whole composite in the loading direction. Variations in textile ratios were achieved by changing the number of textile layers. The BTRC composite was reinforced with one-, two-, three-, and four-layer of basalt, while corresponding textile ratios were 1.22%, 1.44%, 1.96%, and 2.30%, respectively.

Wooden molds with an internal cross-section of 400 mm × 50 mm were used to cast tensile specimens, while the thickness depends on specified textile ratios (Figure 3). The length and width of the specimens were 400 and 50 mm, respectively. As textile layers increased from one to four, the thickness was 12, 20, 22, and 25 mm, respectively. BTRC specimens would be cured at room temperature for 28 days before further tension tests. In total, 21 thin plates were prepared for the tension tests. To distinguish different test setups, the labels C and I were adopted to represent clevis-grip and improved clevis-grip, respectively. Specimens are identified as C/I-LX-#, where X refers to the number of the basalt textile layers, and the digit # indicates the serial number of specimens. Table 2 reports the specimen ID, geometry, test setups, number of textile layers, and textile ratios for better clarification.

### 2.3. Test Setup

In accordance with ACI PRC-549.4-20 guidelines [18], an Instron system with a load capacity of 250 kN was adopted to perform tensile tests under displacement control at a loading rate of 0.2 mm/min (Figure 4). The deformations were measured using a pair of linear variable differential transformers (LVDTs) at the middle region to record relative displacement measurements. In particular, two strain gauges were attached to the middle surfaces of thin plates to measure the deformations of the matrix under external loads before cracking. Four perforated steel plates (dimensions 100 × 50 × 5 mm) were applied using epoxy resin at the extremities of the thin plates, then connected with the testing machine by pins (Figure 4). Therefore, the tensile loads can transfer from the test machine to the steel plates and finally to the matrix through only shearing stress to realize the tensile loading of the BTRC composite. Two hinges were formed between the specimens and the testing machine to suppress the negative effects of possible loading eccentricity and misalignment.

## 3. Results and Discussions

### 3.1. Crack Patterns and Failure Modes

After tensile tests, the cracking patterns of the specimens were re-depicted to facilitate the identification and further comparison. Figure 5a shows the crack patterns of one representative specimen for each series, while Figure 5b provides related characteristic data, i.e., the number and average cracking space of cracks. For thin plates loaded with a clevis-grip device (specimens C-L1/L2/L4), only one transverse crack was observed at the approximately middle zone, and the fiber yarns slid out from the matrix accompanied by fracture of a few fiber filaments. This is related to the too weak textile–matrix interaction to transfer adequate tensile force [10,11,20,23]. These things considered, the BTRC composite exhibited the same crack patterns and failure modes despite increasing textile ratios. Differently, the specimens tested by improved clevis-grip showed multi-cracking behavior in the case of relatively high textile ratios (exceeding one textile layer). Multi-transverse cracks distributed relatively uniformly along the specimens in the loading direction. The delicate interactions between the textile and material matrix account for the multi-cracking behavior [27,31]. By increasing textile ratios from 1.44% to 2.30%, the cracks increased from 5 to 9 and then decreased to 5, and the corresponding average crack spacing decreased from 4.98 cm to 2.99 cm and then increased to 4.40 cm. It was observed that cracks showed the highest density as the composite was reinforced with three layers of textile (textile ratio = 1.96%). The ultimate failure of all BTRC plates resulted from textile rupture.

The failure modes of the BTRC composite changed for different test setups even with the same textile ratio, e.g., C-L4 and I-L4. In the following, the effects of test setups and textile ratios on tensile stress-strain curves and their characteristic parameters were analyzed to understand whether it is appropriate to obtain the design parameters of BTRC using the improved clevis-gripping method and to instruct further engineering applications.

### 3.2. Tensile Stress-Strain Curves and Characteristic Parameters

Figure 6 shows the tensile stress-strain curves for all specimens, while Table 3 reports some key experimental results. Here, stress and strain were obtained by dividing the applied load and relative displacements by the cross-sectional area of the thin plates and gauge length (i.e., 200 mm), respectively.

For the clevis-grip, the tensile stress-strain curves of BTRC can be characterized in two stages: the first stage represents the uncracked state, where the slope of the stress-strain curve reflects the elastic modulus of the matrix. The second stage corresponds to the pull out of the textile from the matrix. It is noteworthy that the strain was measured by strain gauges since LVDTs obtained invalid data due to the minimal deformations. The stress increased linearly with the strain at the beginning of tests in the uncracked stage. A sudden drop then occurred to the tensile load due to the formation of one macro crack. The applied tensile displacement continually increased under further loading, and the fiber yarns slowly slid out from the fine-grained concrete matrix. Despite this, the local deformation obtained by strain gauges decreased owing to the unloading behavior of the matrix. As a supplement, Figure 7 shows the applied tensile load-displacement curves. It can be observed that the curves suddenly dropped after first cracking, followed by a long plateau resulting from fiber yarn pull-out in some cases. The tensile performance of the BTRC composite was governed by the poor textile–matrix interface [19,20]. 

As for the improved clevis-grip, the tensile stress-strain curves showed strain-hardening behavior that can be divided into three stages except for specimens I-L1. In addition, a comparison was made between the mechanical properties of BTRC composite and those of dry textile with an equal amount to better understand the behavior of BTRC composite systems. Therefore, the stress-strain curves of basalt textile are also shown with black dotted lines in the figure. The stress was normalized by the cross-sectional area of specimens; thus, the slope of the curve was taken as the product of the textile elastic modulus and textile ratio. The first nearly linear phase represented an uncracked state up to the first cracking, where the stress-strain curve slope reflects the BTRC composite’s elastic modulus. The second phase corresponds to the formation of the multi-cracking in the concrete matrix. A significant decrease happened to stiffness at this stage, and relatively thin transverse cracks appeared along the length of the specimen. The gradual formation of multi-cracks implied the “pseudo ductility” behavior of the TRC composite. The third stage mainly involved the widening of existing cracks until a rapid load drop related to textile rupture. It can be considered that the textile made a predominant contribution to the tensile resistance, and the stiffness and tensile capacity of the composite material should be equal to those of the dry textile with the same amount. In fact, a fiber yarn consists of hundreds of filaments with diameters of several micrometers, as shown in Figure 2b. The concrete matrix with relatively larger particles can hardly impregnate the full cross-section of the fiber yarns, and a perfect bond only exists for outer filaments (“telescopic behavior”) [32,33,34,35,36,37,38]. During the loading process, the outer fiber filaments are susceptible to premature fracture due to the stress transmitted by the matrix, resulting in a decrease in the effective cross-sectional area of the fiber yarns. Therefore, the stiffness and tensile capacity of the specimen at this stage are lower than the corresponding values of the dry textile. The former phenomenon is termed “stiffness loss”, see Figure 6. The overall response of the BTRC composite was strongly dependent on the mechanical properties of its textile reinforcement.

The cracking and peak stresses were attained at the same point for specimens C-LX. When BTRC composite contained one or two textile layers, the cracking stress ranged between 5.13 MPa and 5.89 MPa, slightly oscillating around the flexural-tension strength of fine-grained concrete. As the textile layers increased to four, the cracking stress dramatically decreased to a range of 2.71 MPa–3.28 MPa. For specimens I-LX, the cracking stress of the BTRC composite was much lower than the flexural-tension strength of fine-grained concrete, ranging from 1 MPa–2 MPa. The BTRC had peak stress between 3.21 MPa and 5.87 MPa. The strain corresponding to peak stress was generally above 1.5%, approaching the maximum strain of fiber yarns (i.e., 1.8%). Furthermore, a coefficient k was defined as the ratio of the tensile capacity of the TRC composite to that of the dry textile with the same amount to characterize the utilization rate of the textile in the TRC systems (Table 3). The range of textile utilization rate was 50–75%.

### 3.3. Effect of Test Setups

Figure 8 compares the force transfer mechanism in TRC tensile tests for different test setups. For the clevis-grip method, the tensile force is transferred from the metallic plates to the fine-grained concrete matrix through interfacial bonding and then transferred to the textile through the matrix-textile interface interactions, realizing the loading of the entire composite. The composites’ tensile response depends on the metallic plates-matrix interface and the matrix-textile interface properties. Significant slippage between textile and matrix occurs at the extremities of the gripping area and along the overall length of the specimen. The obtained results may be unsuitable for characterizing reliable design parameters of the composite owing to their dependencies on the griping systems, such as the length of the gripping part, and on the hardly quantified matrix-textile interface slippage behavior. When the improved clevis-grip method is employed, before first cracking, it can be considered that the tensile force is directly applied to the textile and concrete matrix, and their deformations are synchronous. After cracking, the textile still directly carries the tensile force up to the occurrence of rupture, while the concrete matrix shares a limited part of the tensile force by weak textile–matrix interactions. The obtained results are dependent on more reliable textile properties due to eliminating the textile slippage at the ends of the gripping area (Figure 8b).

Different crack patterns were detected for specimens with the same textile ratio, such as C-L4 and I-L4 (Figure 5). This is attributed to different textile–matrix interface deformation behavior, and the former test setup allows apparent textile slippage while the latter does not. The BTRC composite loaded with improved clevis-grip may provide high tensile resistance at large deformation (Figure 6). Figure 9 demonstrates the effect of test setups on the cracking stress of BTRC composites. The cracking stress significantly decreased after improving the loading method. When the number of textile layers was 1, 2, and 4, the cracking stress decreased by 80.0 %, 69.5 %, and 39.9 %, respectively. The decrease in cracking stress may be caused by larger accident loading eccentricity associated with replacing a stiffer matrix with softer epoxy resin (Figure 1a,c).

### 3.4. Effect of Textile Ratios

Figure 10a shows the effect of textile ratios on the cracking stress of the TRC composite. As the textile ratio increased, the TRC cracking stress of specimens C-LX dropped while that of specimens I-LX gradually increased. For instance, the cracking stress of specimens C-LX decreased by 47.4% while that of specimens I-LX increased by 58.0% as the textile layers increased from 1 to 4. Figure 10b illustrates the effect of the textile ratios on the peak strength, ultimate strain, and textile utilization rate of the specimens tested with improved clevis-grip. All three parameters exhibited a first decrease and subsequent rise by increasing the textile ratios. The strain corresponding to the peak strength of specimens (ultimate strain) was generally less than the maximum textile strain (1.8%). This can be explained by the fact that although the textile mainly resists the tensile force at failure, the concrete matrix still shares part of the tensile force through the interface interactions, thus reducing the composite deformability (matrix stiffening effect). The textile utilization rate was 70.9% when the concrete matrix was incorporated with one textile layer. With the increase in the textile ratios, the utilization rate dramatically decreased and eventually stabilized at nearly 55%. The highest utilization of textile happened to TRC with the lowest textile ratio, which may benefit from fewer damaged outer filaments related to the absence of multi-cracking (Figure 5a).

### 3.5. Modeling of Stress-Strain Behavior

An attempt was made to predict the tensile behavior of TRC derived from improved clevis grip using the ACK model with three stages proposed by Aveston, Cooper, and Kelly [26,27]. The basic assumptions are: (1) the tensile resistance of the textile ratio exceeds the cracking load; (2) the bond between the concrete matrix and the fiber yarns is weak; (3) once debonding between the concrete matrix and fiber yarns occurs, the bonding stress is replaced by friction stress; (4) the frictional stress remains as a constant on the debonding interface. According to the composite mixing law [39], the initial slope, E_1_, of the first linear stage related to the uncracked stage depending on volume fractions and elastic moduli of textile and matrix can be computed as follows:(1)$$E_{1}=E_{f}V_{f}+E_{m}V_{m}$$
where *E_f_* and *E_m_* are elastic moduli of the fiber and mortar matrix, respectively, and *V*_f_ and *V_m_* are the corresponding volume fractions.

The cracking stress, *σ*_1_, and corresponding strain, *ε*_1_, of the TRC composite are given by:(2)$$\sigma_{1}=\frac{\sigma_{mu}}{E_{m}}E_{1}=\sigma_{mu}(V_{m}+\frac{V_{f}}{V_{m}})E_{1}$$
(3)$$\varepsilon_{1}=\frac{\sigma_{1}}{E_{1}}$$

The second stage related to the formation of multi-cracks is considered as a platform, terminating at the strain, *ε*_2_:(4)$$\varepsilon_{2}=\frac{\sigma_{1}}{E_{1}}(1+0.666\frac{E_{m}}{E_{f}}\frac{V_{m}}{V_{f}})\frac{\sigma_{mu}}{E_{m}}$$

The third stage involves a linear branch up to textile rupture, and the stiffness, *E*_2_, peak strength *σ*_u_, and ultimate strain, *ε*_u_, are determined as:(5)$$E_{3}=E_{f}V_{f}$$
(6)$$\sigma_{u}=\sigma_{f}\frac{A_{f}}{A_{c}}=\sigma_{f}V_{f}$$
(7)$$\varepsilon_{u}=\frac{A_{f}\sigma_{f}}{A_{m}E_{3}}-\frac{\sigma_{1}-E_{3}\varepsilon_{2}}{E_{3}}$$
where *A_f_*, *A_m_*, and *A_c_* are the cross-sectional areas of the longitudinal fiber yarns, concrete matrix, and composite, respectively, and *σ*_f_ is the tensile strength of the textile. 

The cracking stress of the TRC matrix was assumed as 1.5 MPa according to the experimental results. Figure 11 shows the comparisons between the experimental tensile stress-strain behavior of TRC and predictions derived from the ACK model. The shaded area indicated variations in the testing results. It can be seen that the ACK model significantly overestimates the slope of the third stage of the stress-strain curves.

If the terminating point of the third branch in the ACK model is modified to consider the utilization rate of the textile as follows:(8)$$\sigma_{u}=0.5\sigma_{f}\frac{A_{f}}{A_{c}}=0.5\sigma_{f}V_{f}$$
(9)$$\varepsilon_{u}=\frac{\sigma_{u}}{E_{f}}$$

That is, the tensile strength of fiber yarns in TRC is only half of that of dry textile, while the ultimate tensile strain is the same as that of dry textile. Then, the modeled tensile stress-strain curves are in good agreement with the experimental curves for all TRC plates.

## 4. Conclusions

The clevis-grip tensile test setup for the BTRC composite was improved by casting epoxy resin at the extremities. This paper primarily reported the experimental results of 21 BTRC thin plates subjected to uniaxial tension with different test setups and textile ratios. The main conclusions can be drawn as follows:(1)After improving the testing method, the failure of the BTRC composite changed from slippage of textile within the matrix to textile rupture; besides, BTRC showed multi-cracking behavior as the textile ratio exceeded 1.44 %.(2)The clevis-grip tensile test is a fiber-matrix bond test rather than a test to characterize the TRC composite. However, the improved clevis-grip tensile test may provide reliable design parameters because the tensile performance of BTRC composite in the cracks widening stage mainly depends on textile properties.(3)For BTRC loaded with the improved clevis-grip tensile test, the peak strength, ultimate strain, and textile utilization rate showed the first drop and a further increase as the textile ratios increased; in addition, BTRC with the lowest textile ratio attained the highest utilization rate of textile.(4)The tensile stress-strain behavior of TRC derived from the improved test method was predicted by the modified ACK model with a textile utilization rate of 50%, showing good agreement.

The key point in this paper is that the proposed tensile test setup for the TRC composite allows for determining reliable design parameters in terms of peak strength, ultimate strain, and cracked modulus. The textile utilization rate plays a crucial role in predicting the tensile stress-strain behavior of the BTRC composite; however, the determination of this parameter is difficult due to limited data. Consequently, more attention should be paid to obtaining the textile utilization rate under the combination of different effects, including fiber type, textile configuration and concrete matrix strength.

## Figures and Tables

**Figure 1 materials-15-08975-f001:**
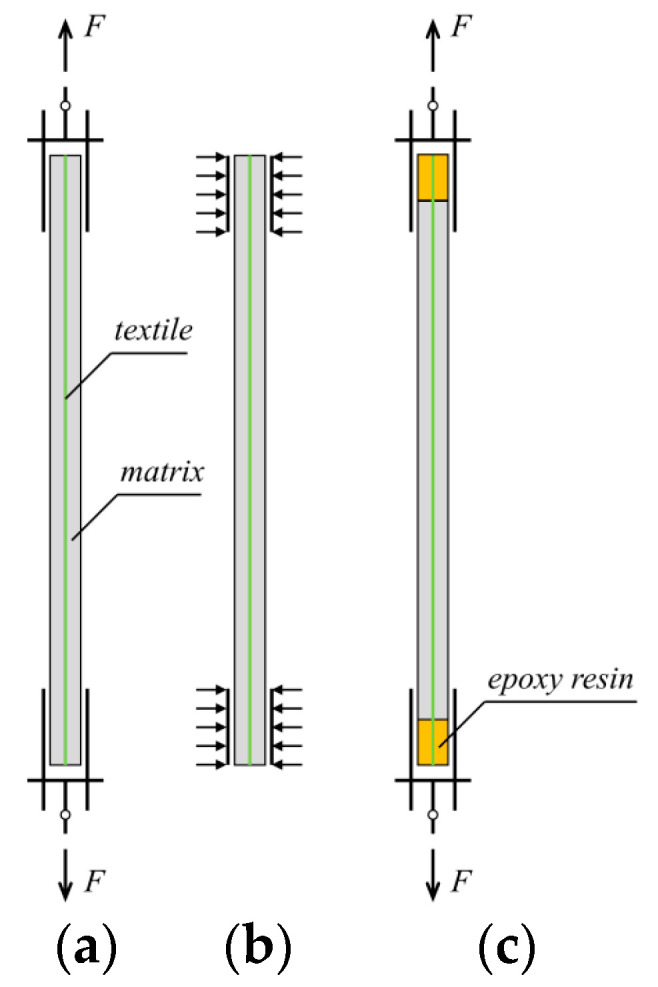
Tensile test setups: (**a**) clevis-grip; (**b**) clamping-grip; and (**c**) improved clevis-grip (proposed by the authors).

**Figure 2 materials-15-08975-f002:**
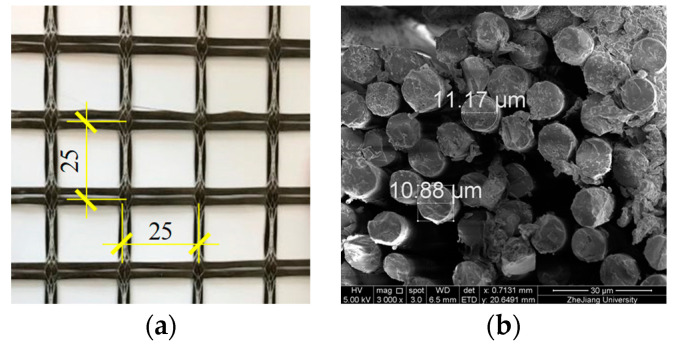
(**a**) the geometry of the textile, mm and (**b**) SEM image of a fiber yarn section.

**Figure 3 materials-15-08975-f003:**
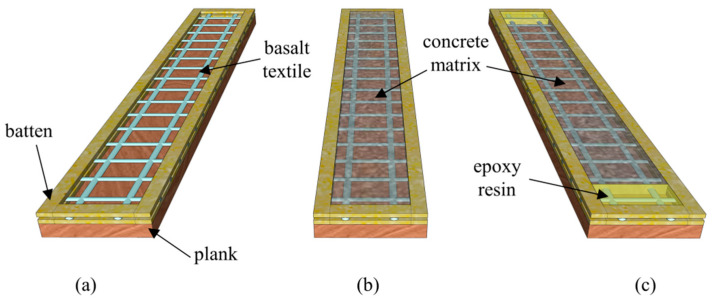
Fabrication of TRC coupons: (**a**) wooden molds; (**b**) casting concrete matrix; and (**c**) casting concrete matrix and epoxy resin in the middle and end zones, respectively.

**Figure 4 materials-15-08975-f004:**
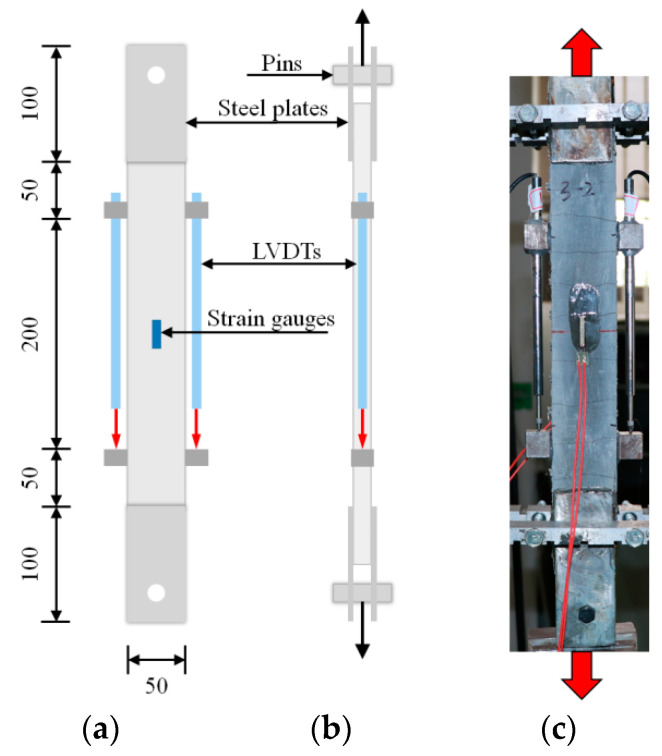
Test setup for tensile tests: (**a**,**b**) sketch, in mm; and (**c**) in-site photos.

**Figure 5 materials-15-08975-f005:**
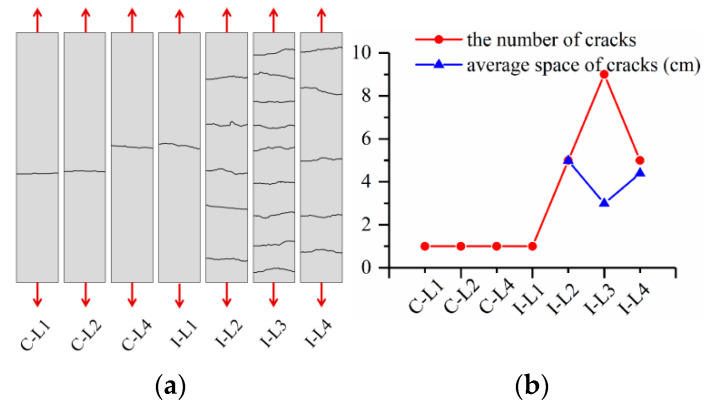
(**a**) crack patterns; and (**b**) number and average spacing of cracks.

**Figure 6 materials-15-08975-f006:**
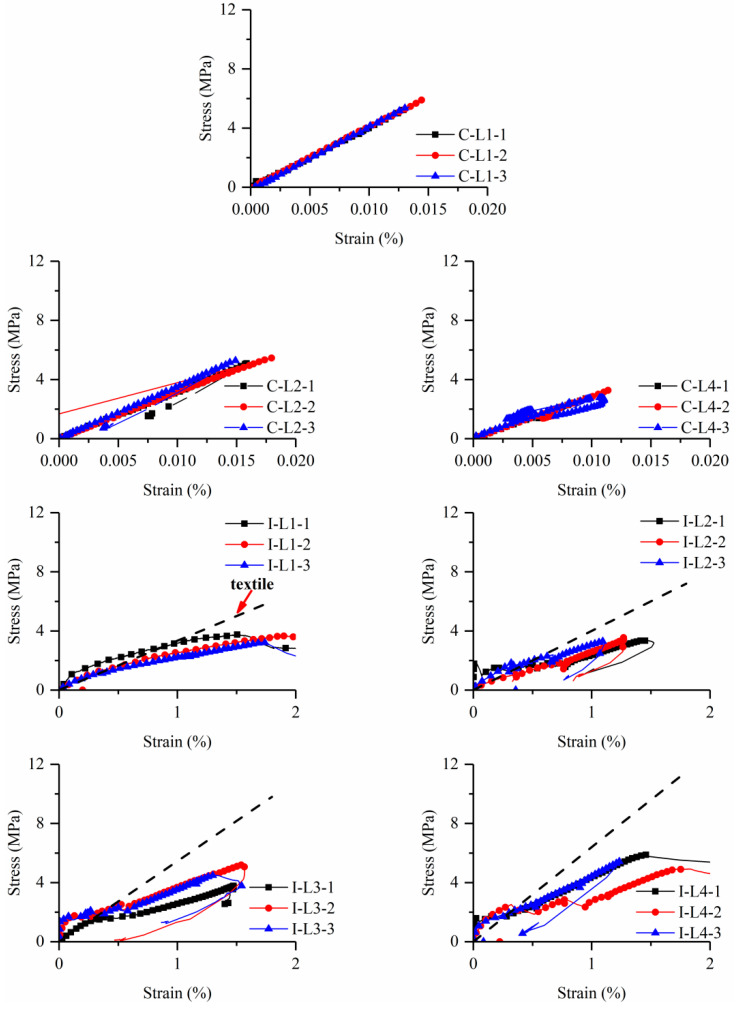
Tensile stress-strain curves of BTRC specimens.

**Figure 7 materials-15-08975-f007:**
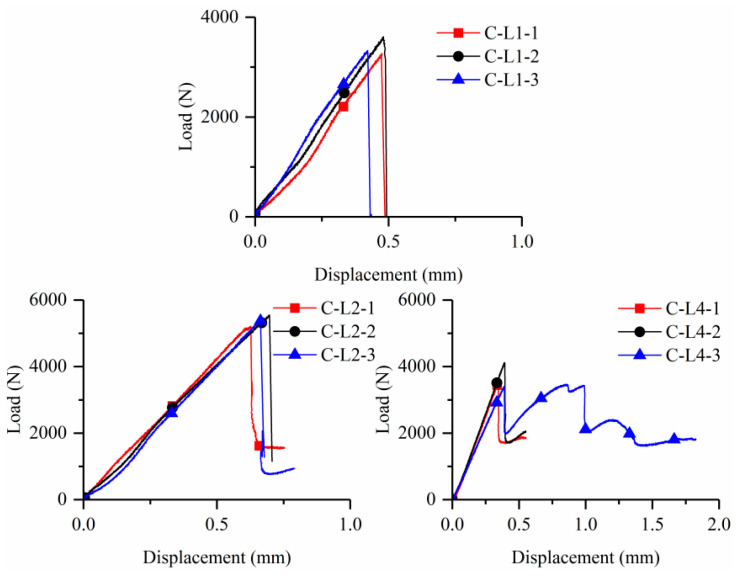
Load-applied displacement curves of BTRC specimens.

**Figure 8 materials-15-08975-f008:**
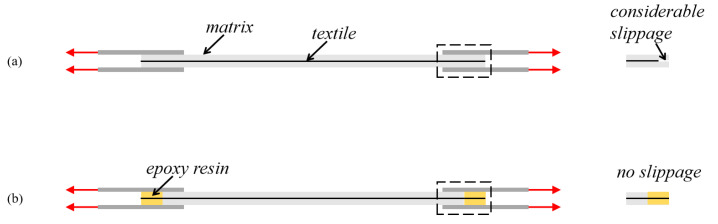
Effect of test setups on force transfer mechanism in TRC composite: (**a**) clevis-grip; (**b**) improved clevis-grip.

**Figure 9 materials-15-08975-f009:**
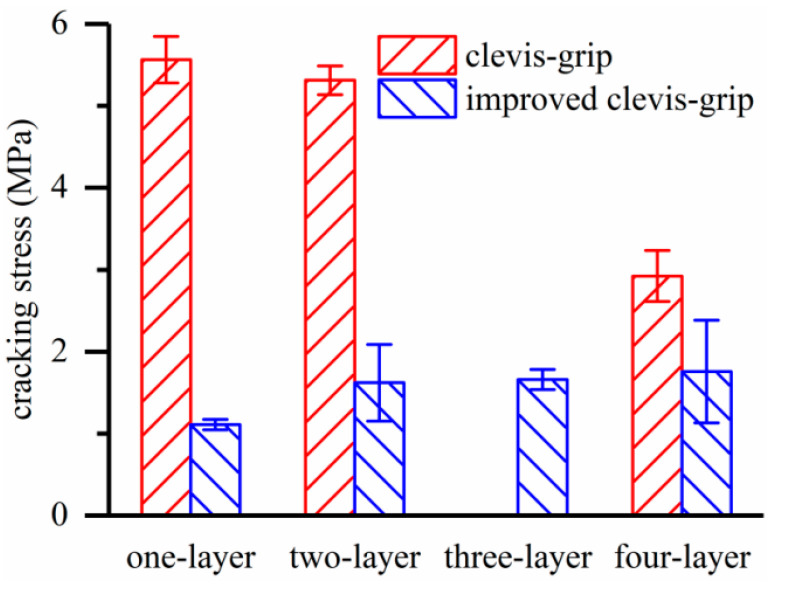
Effect of test setups on cracking stress.

**Figure 10 materials-15-08975-f010:**
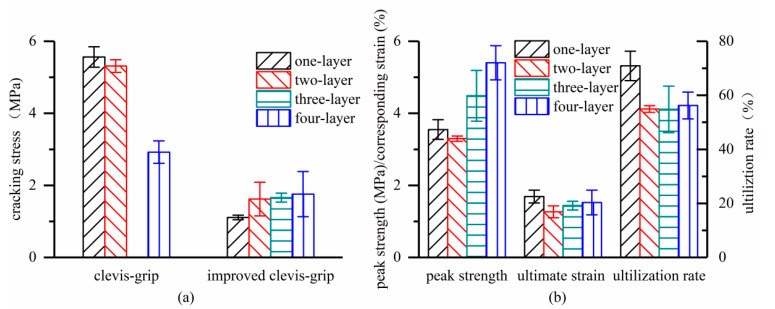
Effect of textile ratios on tensile behavior of TRC: (**a**) cracking stress; and (**b**) peak strength, ultimate strain, and textile utilization rate.

**Figure 11 materials-15-08975-f011:**
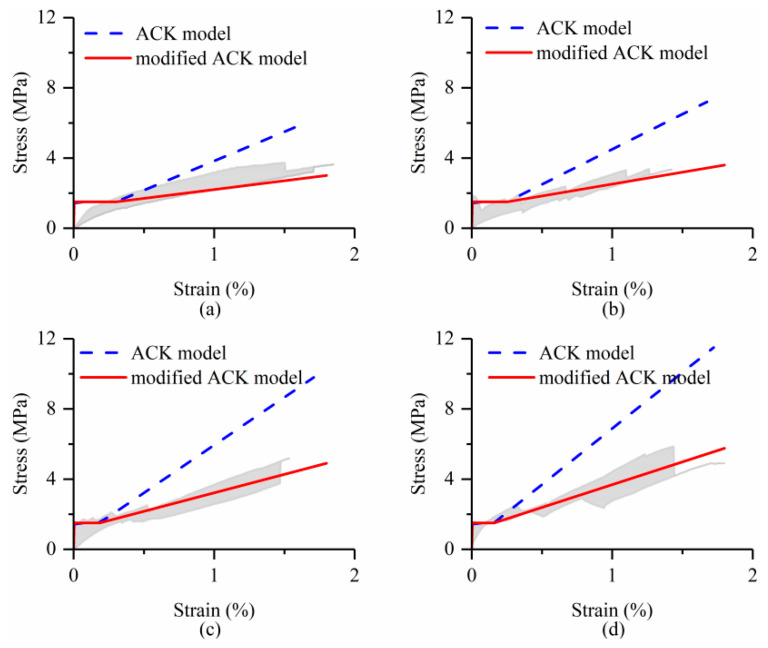
Tensile stress-strain curves of BTRC composites (experimental results VS predictions of ACK model): (**a**) I-L1; (**b**) I-L2; (**c**) I-L3; (**d**) I-L4.

**Table 1 materials-15-08975-t001:** Mix proportion of fine-grained concrete (kg/m^3^).

P·O 52.5 Cement	Fly Ash	Silica Fume	Fine Sand	Superplasticizer	Water
472	168	35	1380	4	240

**Table 2 materials-15-08975-t002:** BTRC thin plates.

Specimen	Thickness (mm)	Width (mm)	Length (mm)	Test Setup	Layers of Basalt Textile	Textile Ratio (%)
C-L1	12	50	400	Clevis-grip	1	1.20
C-L2	20				2	1.44
C-L4	25				4	2.30
I-L1	12			Improved clevis-grip	1	1.20
I-L2	20				2	1.44
I-L3	22				3	1.96
I-L4	25				4	2.30

**Table 3 materials-15-08975-t003:** Experimental results.

Specimen	Cracking Stress σ_cr_/MPa	Cracking Strain ε_cr_/%	Peak Strength σ_u_/MPa	Ultimate Strain ε_u_/%	Textile Utilization Rate k/%
C-L1-1	5.37	0.01326	/	/	/
C-L1-2	5.89	0.01444	/	/	/
C-L1-3	5.43	0.01338	/	/	/
C-L2-1	5.13	0.01596	/	/	/
C-L2-2	5.48	0.01549	/	/	/
C-L2-3	5.33	0.01491	/	/	/
C-L4-1	2.78	0.00986	/	/	/
C-L4-2	3.28	0.01115	/	/	/
C-L4-3	2.71	0.00998	/	/	/
I-L1-1	1.16	0.12809	3.75	1.51	74.9
I-L1-2	1.13	0.27727	3.66	1.86	73.1
I-L1-3	1.04	0.29090	3.24	1.71	64.7
I-L2-1	1.86	0.00832	3.35	1.43	55.8
I-L2-2	1.08	0.21156	3.21	1.27	53.5
I-L2-3	1.92	0.33716	3.33	1.10	55.5
I-L3-1	1.58	0.33222	3.78	1.48	46.2
I-L3-2	1.80	0.13829	5.19	1.54	63.4
I-L3-3	1.60	0.00937	4.49	1.30	54.8
I-L4-1	1.58	0.01928	5.87	1.44	61.1
I-L4-2	2.52	0.32010	4.92	1.90	51.2
I-L4-3	1.33	0.08815	5.42	1.23	56.4

## Data Availability

The raw/processed data required to reproduce these findings cannot be shared at this time as the data also forms part of an ongoing study. The datasets generated during and/or analyzed during the current study are available from the corresponding author upon reasonable request.

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
