# Peer review of "Tensile Behavior of Basalt Textile Reinforced Concrete: Effect of Test Setups and Textile Ratios"

_materials, 2022, doi:10.3390/ma15248975_

Round 1

Reviewer 1 Report

The authors consider an improved set up for studying  BTRC composites materials. 

The paper is clear and well written and the results may be of interest for the composite materials community. 

I just have some minor remarks which need to be taken into account before the paper may be acceptable for publication 

1)line 104: CoVneeds to be more clearly defined. The referee does not know what "coefficient of variation" means

2)line 116:  "areal density of basalt textile was 353 g/m2" areal density should be defined explicitly. What it means is not clear. 

3) Figure 4: the applied force or stress on the samples should be indicated by arrows or any thin appropriate. I count 9 cracks and not 8 in IL-3 sample

Reviewer 2 Report

The present paper is focused on the tensile behavior of basalt textile reinforced concreteThis paper presents the experimental results of tension tests performed on different test setups. The effect of setups and textile ratios on crack patterns, failure mode, and tensile stress-strain curves were investigated in detail. The manuscript research is relevant, and some helpful conclusions are drawn from the study; however, the novelty of the present research is not significant. Overall, the present paper could be published in its original format. before that, the author should modify the abstract and some quantitative results should be added to the abstract.

Reviewer 3 Report

This manuscript evaluates the “Tensile behavior of basalt textile reinforced concrete: effect of test setups and textile ratios”. The manuscript is described and contextualized with the help of previous and present theoretical background. All the references cited are relevant to this area of research. The methods/analytical study are clearly stated. The result and discussion section are clearly presented. The manuscript needs major revision and require the following modifications before the acceptance.

1. Include your research recommendation in the abstract.

2. Keywords: change the keyword Basalt to basalt textile reinforced concrete

3. The introduction need to be elaborated a bit.

4. What is the new perception of your study?

5. Show the images of specimen preparation.

6. The quality of all the graphs need to be enhanced

7. Present the recommendation of your research at the end of the conclusions.

8. Cite your work with more recent relevant literatures.

Round 2

Reviewer 3 Report

The author addressed all the comments well